# Learning to assist smokers through encounters with standardized patients: An innovative training for physicians in an Eastern European country

Narine K. Movsisyan[1,2]*, Varduhi Petrosyan[1], Gohar Abelyan[1], Ondrej Sochor[2], Satenik Baghdasaryan[3], Jean-François Etter[4]

1 American University of Armenia, Gerald and Patricia Turpanjian School of Public Health, Yerevan, Armenia, 2 International Clinical Research Center, St. Anne's University Hospital Brno, Brno, Czech Republic, 3 Yerevan State Medical University, Department of Postgraduate and Continuing Medical Education, Yerevan, Armenia, 4 Institute of Global Health, Faculty of Medicine, University of Geneva, Campus Biotech, Geneva, Switzerland

* narinekm@gmail.com

**Data Availability Statement:** Data cannot be shared publicly because this was not envisioned at the time of applying for the Ethics Committee

## Abstract

### Objectives

A lack of physician training is a major obstacle for effective tobacco dependence treatment. This study assessed the feasibility of an active learning training program and its effects on smoking cessation counselling skills of medical residents in Armenia, an Eastern European country with high smoking prevalence.

### Study design

The study used a pre-post assessment of smoking cessation counselling activities and a course evaluation survey to assess the feasibility of the intervention in a different environment.

### Methods

We adapted an active learning training model developed in Switzerland. Residents were trained in Yerevan, Armenia, using video-taped counselling sessions, role plays, standardized patients (actors), group discussions and immediate feedback. The training evaluation was done using a semi-structured anonymous questionnaire. The study assessed the physicians' self-reported smoking cessation counselling activities before and 6 months after the training. A non-parametric Mann-Whitney test was used to assess pre-post differences in physicians' counselling skills measured on ordinal scale.

### Results

Of the 37 residents trained, 75% were female, 89% aged 20–29 years and 83% were never-smokers. Twenty-eight trainees (76%) returned the course evaluation survey and 32 (86%) answered a questionnaire on skills self-assessment at 6 months follow-up. The majority agreed the course was successful in achieving its learning objectives (64%-96%) and

approval. In particular, the Confidentiality Assurances section of the approved IRB Application had the following statement: "Only the study investigators will have an access to the obtained information as well as the database." The AUA IRB operates under Federalwide Assurance (FWA) for the Protection of Human Subjects Number: FWA00001683. Data requests can be sent to Ms. Varduhi Hayrumyan, the AUA Institutional Review Board's Human Protections Administrator,. E-mail: auairb@aua.am. Information on the AUA IRB is available at https://aua.am/aua-institutional-review-board-guidebook/.

**Funding:** The work of all authors except SB was supported by the Swiss National Science Foundation (SNSF) and the Swiss Agency for Development and Cooperation (SDC), within the SCOPES 2013-2016 Project, SNSF grant IZ73Z0_152616. The SNSF and SDC had no role in the study design, collection, analysis or interpretation of the data, writing the manuscript, or the decision to submit the paper for publication.

**Competing interests:** The authors have declared that no competing interests exist.

increased their confidence in assisting their patients to quit (74%). After 6 months, the physicians were more likely than at baseline to adhere to evidence-based counselling strategies, including assessing the smoking status and dependence and matching the advice to the patient motivation. The training did not, however, improve the prescription of tobacco dependence medications.

## Conclusions

Six months after the training, several self-reported smoking cessation counselling activities had significantly improved compared to baseline. This training model is acceptable for medical residents in Yerevan, Armenia and offers a promising approach in addressing the lack of physician counselling skills in similar settings and populations.

## Introduction

Eastern Europe has the highest rates of smoking in Europe [1] but affordable smoking cessation interventions are lacking in this region [2]. Treatments that combine physician counselling and pharmacotherapy increase smoking cessation rates; such treatments are not just cost-effective, they are cost-saving [3]. Trained physicians are about twice as likely to offer help to patients who smoke as non-trained physicians, and a lack of training or inadequate training is one of the major obstacles for consistent and effective tobacco dependence treatment [4]. The existence of smoking cessation training programs is strongly associated with country income category, to the advantage of high-income countries: only 1% of health care providers trained in smoking cessation in 2007 were working in low-income countries [2]. Another obstacle to the provision of quitting assistance is health care providers' smoking status; physicians who smoke are less likely to advise patients to quit smoking [5]. The majority of European medical students recognize the need for smoking cessation training, but up to 40% of medical students in Eastern European countries use tobacco products [6].

Many postgraduate programs for physicians' smoking cessation training are available in Europe; however, their effectiveness in changing provider practices and patient smoking outcomes is not adequately evaluated. Most often, these courses cover brief intervention, pharmacotherapy, motivational interviewing skills and the stages of behavioural change [7]. One such scheme, an innovative postgraduate training program implemented in teaching hospitals in Switzerland, was shown to improve both physician counselling skills and patient-related outcomes [3]. The effects of this program were explained by the combined impact of active learning methods (as opposed to traditional top-down teaching), the availability of trained faculty and framing the smoking problem in the local (Swiss) context [8]. We recently applied this well-established training program in Armenia, a country in Eastern Europe where 50.9% of men and 3.2% of women are tobacco smokers [1]. In this study, our aim was to assess the feasibility of the training intervention and its effects on the smoking cessation counselling skills of medical residents in Yerevan, Armenia.

## Materials and methods

### Study setting

This training course was conducted at the American University of Armenia (AUA) in Yerevan. The study participants were medical residents of Yerevan State Medical University (YSMU).

## Data collection and instruments

Data for this study were collected from medical residents from May 2015 to January 2016. Data collection included an evaluation survey at the completion of the training course, and a self-assessment of smoking cessation counselling skills before and 6 months after the training course, to allow participants time to practice the skills acquired. All surveys were self-administered, anonymous, and collected by a third party in the absence of the course instructor. The questionnaires validated in previous studies [8] were translated from English in Armenian. The questionnaire that evaluated the cessation counselling course was administered immediately after the training and included 10 items on the course objectives (three-point rating scale), five items to assess the course duration and seven items on the overall quality of the course (five-point Likert scale). The respondents also answered two open-ended questions regarding the strengths and weaknesses of the training course.

The questionnaire that evaluated the medical residents' smoking cessation counselling activities contained 23 items covering tobacco dependence assessment, smoking cessation counselling strategies, and pharmacological treatment. The structured questionnaire had four-point Likert-type response options to assess the residents' use of counselling strategies before and 6 months after the training, including the "5 As": asking about tobacco use, advising every smoker to quit, assessing their willingness to quit, assisting in making a quit attempt and arranging a follow-up visit [9], among others.

## Analysis

Given the purpose of this study was to examine the feasibility of the intervention we aimed to recruit a sample size of 40 participants. A non-parametric Mann-Whitney test was used to assess pre-post changes in physicians' counselling skills measured on an ordinal scale (Likert-type questions). Descriptive statistics were calculated for the overall assessment of the course. Statistical significance was defined at $p < 0.05$.

## Recruitment

The Faculty of Postgraduate and Continuing Medical Education contacted in May-June 2015 a selected sample of medical residents in different years of study at Yerevan State Medical University and invited to the training. The inclusion criteria were the specialization in cardiology, gastroenterology and pulmonology as it was assumed that these specialists were more likely to devote time to smoking cessation counselling. Residents in surgery, gynaecology, emergency medicine and other narrow specialties were not invited. No other restrictions were applied in the recruitment process. All residents meeting the inclusion criteria were invited to the training.

## Training materials

The original Swiss materials, including trainer's manual, case scenarios, slides, video-taped counselling sessions, and patient brochures were translated from French into Armenian by professional services. The Armenian lead co-investigator (first author) reviewed the translated materials for the medical terminology and the relevance to local context and added the data on smoking epidemics in Armenia. In addition, two teaching faculty members of the YSMU Department of Family Medicine reviewed and adjusted the case scenarios and the patient brochures for cultural appropriateness.

## Standardized patient training

Three professional actors were recruited and trained as *standardised patients* using written scenarios featuring various patient profiles across a range of socio-demographic (age, gender, education, marital and employment status) and smoker's characteristics (motivation to quit, addiction severity, etc.).

## Training

A two-day small-group training workshop for medical residents was conducted at the American University of Armenia and included 4-hour sessions scheduled to allow practice between the sessions. In the first session, the trainer briefly introduced the concepts of behaviour change stages in tobacco addiction, followed by a demonstration of video-materials and role plays. The trainees were asked to complete observation checklists to assess the videotaped cases and the role plays. The assessment results were discussed in groups of 3–4 participants and then in a classroom, with a feedback from the trainer. The second day session included a brief presentation on pharmacological treatment options and practicing with standardized patients. Residents received patient education brochures after the training.

# Results

## Participants' characteristics

Of 48 physicians who were invited, 37 attended the training course (77%) and 36 returned the demographic forms. Of these, the majority were female (n = 27, 75%), aged 20–29 years (n = 32, 89%), and non-smokers (n = 30, 83%). Cardiology and gastroenterology were the two most common fields of specialization. Six of nine male physicians were current smokers. Ten physicians, including four females, reported use of other tobacco products, mostly narghile (waterpipe) and cigars (Table 1).

## Training course evaluation

Of 37 trainees, 28 (76%) returned the course evaluation questionnaire. The proportion of participants who believed that the course "completely achieved" its learning objectives varied in a range of 64%-96% depending on the specific objective. The highest level of agreement was on the item "Understanding the physician's role in smoking cessation" followed by "Advise smokers with strategies that match their motivation to quit". Objectives on prescribing medications and increased self-efficacy in helping smokers to quit were met to a lesser extent (Fig 1). The trainees highly appreciated the acquired knowledge and communicating skills. A few found that counselling strategies and video materials could be more relevant to local context.

## Smoking cessation counselling activities

Of 37 participants who completed the baseline self-assessment questionnaire on smoking cessation counselling skills, 32 (86%) answered the same questionnaire at the 6-month follow-up. Significant before-after differences were found on the six of seventeen individual items on counselling strategies (Table 2). The strongest effects were observed on asking about the time to the first cigarette of the day, suggesting an appointment to discuss smoking, and providing self-help materials (all p≤0.001). No differences were found in self-reported medication prescription patterns, including cytisine and nicotine gum/patch.

**Table 1. Participants' characteristics at baseline (n = 36).**

| Demographic characteristics & smoking status | Categories | % (N) |
|---|---|---|
| Gender | Male | 25.0 (9) |
| | Female | 75.0 (27) |
| Age, years | 20–29 | 88.9 (32) |
| | 30–39 | 8.3 (3) |
| | 40–49 | 2.8 (1) |
| Marital status | Single | 69.4 (25) |
| | Married | 27.8 (10) |
| | Divorced/separated | 2.8 (1) |
| Specialty | Cardiology | 44.4 (16) |
| | Gastroenterology | 36.1(13) |
| | Internal medicine | 11.1 (4) |
| | Pulmonology | 5.6 (2) |
| | Dentistry | 2.8 (1) |
| Smoking status | Current smoker | 16.7 (6) |
| | Ex-smoker | 0 (0) |
| | Never-smoker | 83.3 (30) |
| Use of other tobacco products | Narghile | 30.0 (3) |
| | Cigar | 30.0 (3) |
| | Narghile +cigar | 20.0 (2) |
| | Narghile + e-cigarette | 10.0 (1) |
| | Narghile + cigar + e-cigarette | 10.0 (1) |

## Discussion

In this study, we assessed the feasibility of an innovative smoking cessation training course for medical residents in Armenia. Most participants agreed that the training met its objectives to build the smoking cessation counselling skills, was understandable and acceptable. The strongest effects of the training were related to reconsidering the professional role in helping patients to quit and to learning to advise according to patient motivation to quit. In contrast, prescription of pharmacological treatment was less influenced by the training.

Six months after the course, the trained physicians were more likely to report adhering to evidence-based counselling strategies, including offering help, assessing level of dependence, advising on useful behavioural techniques, proposing an appointment to discuss smoking, and providing educational materials. Prescription of pharmacotherapy remained low at follow-up reflecting the course focus on counselling skills and, possibly, the external factors, including a low access to tobacco dependence medications in Armenia, of which only cytisine has been consistently available locally. In addition to the cost and availability of medications, patient characteristics and physician attitudes and beliefs about the effectiveness of medication and counselling may also influence prescribing patterns [10]. The barriers to using tobacco dependence medications need a further exploration in this local context.

In previous reports, the effects of smoking cessation counselling training for medical residents were mixed, from no effect to a significant improvement in some, but not all, counselling activities [11]. The training model developed in Swiss teaching hospitals (adapted and tested in this study) effectively improved the residents' counselling skills but not their prescribing practices [3]. Similarly, a Cochrane meta-analysis found a measurable effect of the training

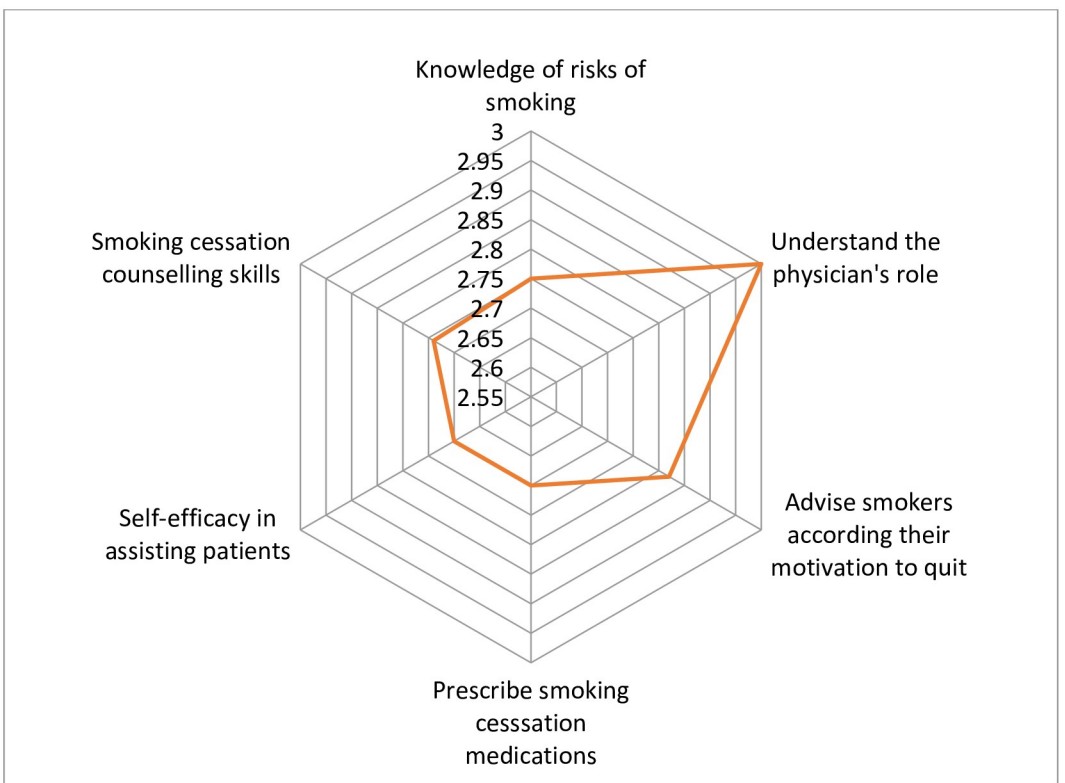

**Fig 1. Achievement of the training objectives.** Each axis of the spider graph represents one of the objectives. The scale on each axis represents the level of achievement on a Likert-like scale (not achieved at all, partially achieved, completely achieved).

on professional performance and on patients' cessation rates but not on provision of tobacco dependence treatments, specifically nicotine replacement therapy [12]. This Cochrane review was, however, based on studies performed in high-income countries such as the US, UK, and Switzerland.

Our study adds to the scarce evidence on the medical residents' smoking cessation training in Eastern Europe, a region with high smoking rates. To our knowledge, this is the first attempt to apply active learning methods, including standardized patients, to train physicians on tobacco dependence treatment and counselling in post-soviet countries. The ample follow-up period, the high response to the follow-up survey, the use of anonymous and validated questionnaires, and the focus on physician behaviour (vs. knowledge and attitude) are the advantages of the study. The study assessed only self-reported cessation counselling activities, as objective verification was not feasible in this context. Focusing on physician behaviour rather than both physician and patient-related outcomes is another limitation of this feasibility study. Besides, the participants represented only selected fields of specialization whereas other specialists may have a different level of interest in smoking cessation counselling training. No control group was used in this feasibility study with a pre-post design; therefore, the effectiveness of the training program cannot be determined.

Future studies will aim to evaluate the effectiveness of the training model in changing provider and patient related outcomes and to explore the ways to further improve physician counselling and prescribing skills to reach out to the smoker patients in restricted-resource settings.

**Table 2. Physicians' self-reported counselling activities before and after the training course.**

| Counselling strategy | Before % (n)(N = 37) | | | | After % (n)(N = 32) | | | | P*-value |
|---|---|---|---|---|---|---|---|---|---|
| | Always | Often | Sometimes | Never | Always | Often | Sometimes | Never | |
| Ask if the patient smokes | 64.9 (24) | 24.3 (9) | 10.8 (4) | 0.00 (0) | 78.3 (25) | 18.7 (6) | 3.13 (1) | 0.0 (0) | 0.193 |
| Ask about number of cigarettes per day | 46.0 (17) | 32.4 (12) | 18.9 (7) | 2.7 (1) | 53.1 (17) | 25.0 (17) | 21.9 (7) | 0.0 (0) | 0.638 |
| Ask about time to first cigarette of the day | 2.7 (1) | 2.7 (1) | 29.7 (11) | 64.9 (24) | 18.8 (6) | 21.9 (7) | 40.6 (13) | 18.8 (6) | **<0.001** |
| Ask if patient smokes indoors at home | 0.00 (0) | 10.8 (4) | 29.7 (11) | 59.5 (22) | 15.6 (5) | 12.5 (4) | 46.9 (15) | 25.0 (8) | **0.003** |
| Ask if patient intends to stop smoking | 40.5 (15) | 40.5 (15) | 15.2 (6) | 2.7 (1) | 59.4 (19) | 37.5 (12) | 3.1 (1) | 0.00 (0) | 0.052 |
| Advise reducing the number of daily cigarettes | 61.1 (22) | 22.2 (8) | 11.1 (4) | 5.6 (2) | 56.3 (18) | 25.0 (8) | 12.5 (4) | 6.3 (2) | 0.670 |
| Advise stopping smoking abruptly | 73.0 (27) | 13.5 (5) | 10.8 (4) | 2.7 (1) | 50.0 (16) | 28.1 (9) | 21.9 (7) | 0.00 (0) | 0.076 |
| Discuss the health risks of smoking | 48.7 (18) | 40.5 (15) | 8.1 (3) | 2.7 (1) | 46.9 (15) | 28.1 (9) | 25.0 (8) | 0.00 (0) | 0.525 |
| Discuss the benefits of smoking cessation | 46.0 (17) | 29.7 (11) | 18.9 (7) | 5.4 (2) | 46.9 (15) | 25.0 (8) | 29.1(9) | 0.00 (0) | 0.990 |
| Discuss personal barriers to cessation | 13.5 (5) | 29.7 (11) | 37.8 (14) | 18.9 (7) | 15.6 (5) | 28.1 (9) | 43.8(14) | 12.5 (4) | 0.713 |
| Propose helping the patient quit | 22.2 (8) | 8.3 (3) | 27.8 (10) | 41.7 (15) | 28.1 (9) | 25.0 (8) | 37.5 (12) | 9.4 (3) | **0.017** |
| Advise on behavioural "tricks" | 10.8 (4) | 13.5 (5) | 27.0 (10) | 48.7 (18) | 9.4 (3) | 25.0 (8) | 46.9 (15) | 18.8 (6) | **0.048** |
| Give practical advice to prevent relapse | 2.7 (1) | 35.1 (13) | 27.0 (10) | 35.1 (13) | 12.5 (4) | 28.1 (9) | 46.9 (15) | 12.5 (4) | 0.144 |
| Give self-help materials | 2.7 (1) | 10.8 (4) | 40.5 (15) | 46.0 (17) | 9.4 (3) | 43.8 (14) | 40.6 (13) | 6.3 (2) | **<0.001** |
| Propose an appointment to discuss smoking | 0.00 (0) | 2.7 (1) | 16.2 (6) | 81.1 (30) | 6.3 (2) | 12.5 (4) | 37.5 (12) | 43.8 (14) | **0.001** |
| Refer patient to other specialists | 0.00 (0) | 5.6 (2) | 19.4 (7) | 75.0 (27) | 3.1 (1) | 6.3 (2) | 31.3 (10) | 59.4 (19) | 0.171 |
| Suggest setting a specific date for quitting smoking | 5.4 (2) | 16.2 (6) | 10.8 (4) | 67.6 (25) | 12.5 (4) | 9.4 (3) | 37.5 (12 | 40.6 (13) | 0.074 |
| Prescribe cytisine (Tabex®) | 2.7 (1) | 2.7 (1) | 10.8 (4) | 83.8 (31) | 0.00 (0) | 9.4 (3) | 28.1 (9) | 62.5 (20) | 0.057 |
| Prescribe NRT (gum) | 0.00 (0) | 0.00 (0) | 24.3 (9) | 75.7 (28) | 0.00 (0) | 9.4 (3) | 28.1 (9) | 62.5 (20) | 0.170 |
| Prescribe NRT (patch) | 0.00 (0) | 2.8 (1) | 8.3 (3) | 88.9 (32) | 0.00 (0) | 0.00 (0) | 28.1 (9) | 71.9 (23) | 0.092 |
| Prescribe any medication (patch or gum or cytisine) | 2.7 (1) | 5.4 (2) | 24.3 (9) | 67.6 (25) | 0.00 (0) | 12.5 (4) | 34.4 (11) | 53.1 (17) | 0.237 |

*P-value, Mann-Whitney test

## Ethics approval

Approval for the study was obtained from the Institutional Research Board (IRB) of the American University of Armenia (AUA-2014-027). All study participants provided oral informed consent before entering the study.

## Supporting information

**S1 File.**
(DOCX)

**S2 File.**
(DOCX)

## Acknowledgments

We thank Dr. Jean-Paul Humair for kindly sharing the smoking cessation course materials and for receiving NM and OS to a training session with standardized patients at the University of Geneva Hospital.

## Author Contributions

**Conceptualization:** Narine K. Movsisyan, Varduhi Petrosyan, Ondrej Sochor, Jean-François Etter.

**Data curation:** Gohar Abelyan, Satenik Baghdasaryan.

**Formal analysis:** Narine K. Movsisyan.

**Funding acquisition:** Narine K. Movsisyan, Varduhi Petrosyan, Ondrej Sochor, Jean-François Etter.

**Investigation:** Narine K. Movsisyan.

**Methodology:** Narine K. Movsisyan, Varduhi Petrosyan, Ondrej Sochor, Jean-François Etter.

**Project administration:** Narine K. Movsisyan, Varduhi Petrosyan.

**Resources:** Satenik Baghdasaryan.

**Supervision:** Jean-François Etter.

**Writing – original draft:** Narine K. Movsisyan.

**Writing – review & editing:** Narine K. Movsisyan, Varduhi Petrosyan, Gohar Abelyan, Ondrej Sochor, Satenik Baghdasaryan, Jean-François Etter.

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
