## [Decision Letter · Decision Letter 0]

16 Aug 2019

PONE-D-19-16171

Learning to assist smokers through encounters with standardized patients: an innovative training for physicians in an Eastern European country

PLOS ONE

Dear Dr Movsisyan,

Thank you for submitting your manuscript to PLOS ONE. After careful consideration, we feel that it has merit but does not fully meet PLOS ONE’s publication criteria as it currently stands. Therefore, we invite you to submit a revised version of the manuscript that addresses the points raised during the review process.

The paper concisely presents the findings of a feasibility study. It is generally clear and readable. Please refer to my comments and those of the reviewer to make minor amendments to the paper.

Throughout – please check your reference formatting – need a space before the (citation).

Page 3 ‘one such scheme’ – it is unclear what this refers to. I suggest breaking the introduction into several paragraphs and beginning a new one at this point – introduce the concept of various programs designed to improve physicians smoking cessation skills and provide a few examples of other programs – not just the Swiss example you have adapted.

Please clarify and emphasise this as a feasibility study and note the future need to evaluate effectiveness through evaluation of practitioner behaviour.

We would appreciate receiving your revised manuscript by Sep 30 2019 11:59PM. To enhance the reproducibility of your results, we recommend that if applicable you deposit your laboratory protocols in protocols.io, where a protocol can be assigned its own identifier (DOI) such that it can be cited independently in the future. For instructions see: http://journals.plos.org/plosone/s/submission-guidelines#loc-laboratory-protocols

We look forward to receiving your revised manuscript.

Kind regards,

Conor Gilligan

Academic Editor

PLOS ONE

Journal Requirements:

4. Please correct your reference to "p=0.000" to "p<0.001" or as similarly appropriate, as p values cannot equal zero.

Additional Editor Comments:

The paper concisely presents the findings of a feasibility study. It is generally clear and readable. Please refer to my comments and those of the reviewer to make minor amendments to the paper.

Throughout – please check your reference formatting – need a space before the (citation).

Page 3 ‘one such scheme’ – it is unclear what this refers to. I suggest breaking the introduction into several paragraphs and beginning a new one at this point – introduce the concept of various programs designed to improve physicians smoking cessation skills and provide a few examples of other programs – not just the Swiss example you have adapted.

Please clarify and emphasise this as a feasibility study and note the future need to evaluate effectiveness through evaluation of practitioner behaviour.

Reviewers' comments:

Reviewer's Responses to Questions

**Comments to the Author**

1. Is the manuscript technically sound, and do the data support the conclusions?

Reviewer #1: Yes

2. Has the statistical analysis been performed appropriately and rigorously? 

Reviewer #1: Yes

3. Have the authors made all data underlying the findings in their manuscript fully available?

Reviewer #1: Yes

4. Is the manuscript presented in an intelligible fashion and written in standard English?

Reviewer #1: Yes

5. Review Comments to the Author

Reviewer #1: The revised version of this manuscript addresses the previous issues raised. Below are some minor additional suggestions.

1) In the discussion could consider acknowledging that the design was a pre-post design for this feasibility study and given that there was no control group the effectiveness of the training program cannot be determined.

2) In the analysis section could consider rephrasing the first sentence to "Given the purpose of this study was to examine feasibility we aimed to recruit a sample size of 40 participants."

6. PLOS authors have the option to publish the peer review history of their article (what does this mean?). If published, this will include your full peer review and any attached files.

Reviewer #1: No

---

## [Author Response · Author response to Decision Letter 0]

28 Aug 2019

August 19, 2019

Dr. Conor Gilligan

Academic Editor

PLOS ONE

PONE-D-19-16171

Learning to assist smokers through encounters with standardized patients: an innovative training for physicians in an Eastern European country

Dear Dr. Gilligan,

Thank you very much for the review of our manuscript "Learning to assist smokers through encounters with standardized patients: an innovative training for physicians in an Eastern European country" and the valuable feedback. 

On behalf of the authors’ team, I am pleased to submit a revised manuscript with addressed comments raised by the editorial team during the review process. Please kindly find the point-by-point description of our responses to the editors’ and reviewer’s comments. 

Thank you very much for your consideration. 

Sincerely,

Narine Movsisyan, MD, MPH

Senior Researcher

School of Public Health

American University of Armenia 

40 Baghramyan Avenue 

Yerevan 0019 Armenia

Phone: +374 60 612 592

Responses to the Editor’s comments

Editor’s comments:

1. Please check your reference formatting – need a space before the (citation).

Response: The formatting was corrected throughout the text.

2. Page 3 ‘one such scheme’ – it is unclear what this refers to. I suggest breaking the introduction into several paragraphs and beginning a new one at this point – introduce the concept of various programs designed to improve physicians smoking cessation skills and provide a few examples of other programs – not just the Swiss example you have adapted.

Response: Thank you very much for the suggestion. We started a new paragraph with more information on the smoking cessation graduate training programs available in Europe to put the Swiss program into the context. 

“Many postgraduate programs for physicians’ smoking cessation training are available in Europe; however, their effectiveness in changing provider practices and patient smoking outcomes is not adequately evaluated. Most often, these courses cover brief intervention, pharmacotherapy, motivational interviewing skills and the stages of behavioural change (7). One such scheme, an innovative postgraduate training program implemented in teaching hospitals in Switzerland, was shown to improve both physician counselling skills and patient-related outcomes (3).”

3. Please clarify and emphasise this as a feasibility study and note the future need to evaluate effectiveness through evaluation of practitioner behaviour. 

Response: We now clearly state that this is a feasibility study and that future studies would need to employ a more rigorous design for the effectiveness evaluation. 

“No control group was used in this feasibility study with a pre-post design; therefore, the effectiveness of the training program cannot be determined.

Future studies will aim to evaluate the effectiveness of the training model in changing provider and patient related outcomes and to explore the ways to further improve physician counselling and prescribing skills to reach out to the smoker patients in restricted-resource settings.” 

4. Please correct your reference to "p=0.000" to "p<0.001" or as similarly appropriate, as p values cannot equal zero.

Response: Done.

Responses to the Reviewer’s comments

Reviewer’s comments:

The revised version of this manuscript addresses the previous issues raised. Below are some minor additional suggestions.

1. In the discussion could consider acknowledging that the design was a pre-post design for this feasibility study and given that there was no control group the effectiveness of the training program cannot be determined.

Response: Thank you for your suggestion. We have added a clear statement on this important study limitation.

“No control group was used in this feasibility study with a pre-post design; therefore, the effectiveness of the training program cannot be determined”.

2. In the analysis section could consider rephrasing the first sentence to "Given the purpose of this study was to examine feasibility we aimed to recruit a sample size of 40 participants."

Response: The justification for the study sample size was re-phrased as suggested. 

“Given the purpose of this study was to examine the feasibility of the intervention we aimed to recruit a sample size of 40 participants.”

Other points: A legend was added in the Figure 1 as requested.

---

## [Editor Report · Decision Letter 1]

9 Sep 2019

Learning to assist smokers through encounters with standardized patients: an innovative training for physicians in an Eastern European country

PONE-D-19-16171R1

Dear Dr. Movsisyan,

We are pleased to inform you that your manuscript has been judged scientifically suitable for publication and will be formally accepted for publication once it complies with all outstanding technical requirements.

With kind regards,

Conor Gilligan

Academic Editor

PLOS ONE

Additional Editor Comments (optional):

Thank you for carefully addressing the comments raised about your paper. The paper reads well and makes an interesting and worthwhile contribution to the literature.
---

## [Editor Report · Acceptance letter]

16 Sep 2019

PONE-D-19-16171R1 

Learning to assist smokers through encounters with standardized patients: an innovative training for physicians in an Eastern European country 

Dear Dr. Movsisyan:

I am pleased to inform you that your manuscript has been deemed suitable for publication in PLOS ONE. Congratulations! Your manuscript is now with our production department. 

With kind regards,

on behalf of

Dr. Conor Gilligan 

Academic Editor

PLOS ONE